# Reactivation of Varicella-Zoster Virus in Patients with Lung Cancer Receiving Immune Checkpoint Inhibitors: Retrospective Nationwide Population-Based Cohort Study from South Korea

**DOI:** 10.3390/cancers16081499

**Published:** 2024-04-14

**Authors:** Jiyun Jung, Seong-Yeon Park, Jae-Yoon Park, Dalyong Kim, Kyoungmin Lee, Sungim Choi

**Affiliations:** 1Clinical Trial Center, Dongguk University Ilsan Hospital, Goyang 10326, Republic of Korea; bestjudy21@gmail.com; 2Research Center for Chronic Disease and Environmental Medicine, Dongguk University College of Medicine, Gyeongju 38066, Republic of Korea; nephrojyp@gmail.com; 3Division of Infectious Diseases, Department of Internal Medicine, Dongguk University Ilsan Hospital, Goyang 10326, Republic of Korea; psy99ch@hanmail.net; 4Department of Internal Medicine, College of Medicine, Dongguk University, Gyeongju 38066, Republic of Korea; moondragon81@naver.com; 5Division of Nephrology, Department of Internal Medicine, Dongguk University Ilsan Hospital, Goyang 10326, Republic of Korea; 6Division of Hematology and Medical Oncology, Department of Internal Medicine, Dongguk University Ilsan Hospital, Goyang 10326, Republic of Korea; 7Division of Hemato-Oncology, Department of Internal Medicine, Korea University Guro Hospital, Korea University College of Medicine, Seoul 08308, Republic of Korea; brightsky88@korea.ac.kr

**Keywords:** herpes zoster, immune checkpoint inhibitors, standardized incidence ratio, cancer patients

## Abstract

**Simple Summary:**

This study investigated the association between immune checkpoint inhibitors (ICIs) and risk of herpes zoster (HZ) in lung cancer patients. Analyzing data from 51,021 South Korean patients, it was found that the incidence of HZ in the ICIs group was lower compared to the non-ICIs group. Additionally, ICIs were associated with a 31% reduction in HZ risk, particularly notable among females and those under 68 years old. These findings suggest that ICIs treatment might decrease the risk of HZ in lung cancer patients, and provide valuable insights prompting further research into ICIs influence on infectious diseases.

**Abstract:**

Background: This study aimed to determine the association between immune checkpoint inhibitors (ICIs) and the risk of herpes zoster (HZ) incidence in patients with lung cancer. Method: We obtained national claims data of 51,021 patients from South Korea with lung cancer between August 2017 and December 2021. The study population was classified into ICI and non-ICI groups based on the prescription of ICIs at least once during the study period. To estimate the effects of ICIs treatment compared with those without ICIs treatment on HZ incidence, we used the Cox proportional hazards model adjusted for sex, age, comorbidities, and concomitant use of immunosuppressive drugs. Stratified analyses based on sex, age, and comorbidities were conducted to identify corresponding risk factors. Results: Of the 51,021 study participants, 897 (1.8%) were prescribed ICIs and 2262 (4.4%) were diagnosed with HZ. Approximately 75.6% of the patients receiving ICIs were male, and the prevalence of diabetes, cardiovascular disease, and chronic lung disease in the ICI group was significantly lower than that in the non-ICIs group. The Kaplan–Meier plot showed that the probability of incidence of HZ in the ICIs group was lower than that in the non-ICIs group. Additionally, treatment with ICIs was associated with a 31% lower incidence of developing HZ when compared to that seen without ICIs treatment (95% confidence interval [CI], 0.48–1.00). This association was stronger in females (hazard ratio [HR], 0.42; 95% CI, 0.19–0.94) and those less than 68 years of age (HR, 0.58; 95% CI, 0.34–0.99). Conclusions: In these real-world data from an Asian population with lung cancer, ICIs treatment might be associated with a reduced risk of HZ compared to that without ICIs treatment.

## 1. Introduction

Primary infection with varicella-zoster virus (VZV) results in chickenpox. Herpes zoster (HZ), commonly known as shingles, occurs when VZV reactivates from its latent state within the cranial nerve or dorsal-root ganglia. This reactivation leads to the virus spreading along the sensory nerve to the dermatome, resulting in the manifestation of a painful vesicular rash [1]. In the general population, the incidence of HZ is 2 to 3 per 1000 patients per year, and the lifetime risk is approximately 30% [2].

Reactivation of VZV is associated with a decline in cell-mediated immunity (CMI) against it [3], either as a natural consequence of aging or due to immunosuppression [4]. The major risk factor for the development of HZ is old age, with an odds ratio (OR) of 1.20 (1.10–1.31) per five-year interval in individuals more than 65 years of age [5]. Immunocompromised individuals, including recipients of solid organ or hematopoietic stem cell transplants, those receiving chemotherapy for malignancies, and those with human immunodeficiency virus (HIV) infections, are also at an increased risk of HZ. Studies have shown that individuals receiving cancer treatment have a higher likelihood of experiencing HZ along with more severe symptoms and complications, including disseminated skin lesions, pneumonia, meningitis, and hepatitis, compared to the general population [6,7,8].

Cytotoxic chemotherapy and targeted therapies have been conventional approaches in cancer treatment. However, in recent years, there has been a significant breakthrough in medical oncology with the introduction of immune checkpoint inhibitors (ICIs). These antibodies target molecules like cytotoxic T-lymphocyte antigen 4 (CTLA-4), programmed death receptor 1 (PD-1), and programmed death ligand 1 (PD-L1), ushering in a new era in the treatment of various types of cancers [9]. Unlike conventional chemotherapy, which directly targets cancer cells to induce cell death, ICIs work by blocking immune checkpoint pathways involved in immune system regulation, thus enhancing the body’s immune response against cancer cells [10]. For example, PD-1 and its ligands play negative regulatory roles in the immune response. Consequently, anti-PD-1 antibodies are regarded as immunostimulatory agents that reactivate anergic cytotoxic T cells induced by tumor cells. As a result, ICIs are not expected to cause immunosuppression, and the potential risk of infectious diseases related to ICIs has received little attention from medical professionals.

Limited studies have been performed on the impact of ICIs on the reactivation of VZV, which is influenced by decreased CMI, specifically with regard to T-cell immunity. Published studies primarily consist of sporadic case reports [11,12,13] and retrospective analyses with small patient cohorts from a single center [14].

Therefore, assuming that ICI treatment may affect the risk of developing HZ, we aimed to evaluate the incidence of HZ in patients with lung cancer receiving ICIs (ICI group), such as PD-1 antibodies (nivolumab and pembrolizumab) or PD-L1 antibodies (atezolizumab and durvalumab), and compared it with that in patients receiving cytotoxic chemotherapeutic agents and/or targeted therapy (non-ICI group) using real-world data. To investigate this relationship, we estimated the incidence of HZ in the ICI and non-ICI groups using the standardized incidence ratio (SIR) and we used a Cox proportional hazards model for data of patients with total and subgroup lung cancers, which was obtained from a nationwide claims database in South Korea.

## 2. Methods

### 2.1. Study Design, Database, and Population

For this population-based retrospective cohort study, we used data from a nationwide claims database in South Korea (assigned number: Health Insurance Review and Assessment (HIRA) Research Data (M20220920004)). The National Health Insurance (NHI) system is a nonprofit health insurance program in South Korea. It serves as the central authority responsible for assessing and approving medical service fee coverage for the entire population, except for recipients of medical aid (3%), thereby ensuring high-quality healthcare. Healthcare providers are required to submit claims for reimbursement, which are then collected and assessed by the HIRA Service. The HIRA database provides de-identified demographic and clinical information related to submitted medical fees, general details with respect to patient specifications (age, sex, department, date of diagnosis, primary and secondary diagnoses, date of admission, and hospital arrival pathway), healthcare services (service category, drug codes, daily dosages, days and quantity of supply, and unit price), diagnosis information (main and sub-diagnosis), and details of outpatient prescriptions (drug codes, daily dosages, days and quantity of supply, and unit price).

In South Korea, NHI coverage for ICIs has been approved for patients with advanced non-small-cell lung cancer (NSCLC) with stage IIIB or higher and who were previously treated using platinum-containing chemotherapy, including the following: (1) nivolumab and pembrolizumab since August 2017, (2) atezolizumab since January 2018, and (3) durvalumab since April 2020, according to the PD-L1 expression rate. Considering the NHI coverage period, we obtained the national claims data of patients with lung cancer who were older than 18 years between August 2017 and December 2021. Adult patients with lung cancer were identified using the International Classification of Disease-10 (ICD-10) code C34 for primary or secondary diagnoses.

We excluded solid organ transplant patients (ICD-10 code, Z94); concealed data, including mental illness, rare diseases, sexually transmitted infections, and HIV; patients with hematologic malignancy (ICD-10 code, C81–96); patients who were not given chemotherapy; and patients who were prescribed two or more ICIs simultaneously. Additionally, patients treated with a combination of ICIs and untreated individuals were excluded to identify the effects of the ICIs.

### 2.2. Definitions and Outcomes

The ICI-treated group included patients who were prescribed ICIs at least once after the diagnosis of lung cancer. Patients without a history of ICI prescription during the study period were considered as the non-ICI group. We defined the baseline as the date of first prescription of ICIs or as no prescription (non-ICIs after the diagnosis of lung cancer). The main outcome of interest was the diagnosis of HZ. New cases of HZ were identified by the presence of relevant diagnostic codes for HZ (ICD-10 code B02) in the claims data included in this study after the baseline. Additionally, in the development of HZ, the following potential confounding factors were identified based on claims records between the date of lung cancer diagnosis and the first prescription date of ICIs or non-prescription of ICIs: age; sex; diabetes; cardiovascular disease; chronic lung disease; chronic kidney disease; chronic liver disease; rheumatic disease; and concomitant use of immunosuppressants or steroids. Corticosteroid use was defined as the presence of prescription records for prednisone equivalents ≥ 15 mg/day for at least 14 days, with a prescription history during the study period or the preceding 12 months. All drug codes for ICIs, non-ICIs, corticosteroids, and immunosuppressants as well as ICD-10 codes for comorbidities are listed in Appendix A.

### 2.3. Statistical Analyses

Categorical data for the 10-year age groups, sex, comorbidities, and concomitant use of immunosuppressive drugs were compared using the χ^2^ test. Additionally, we identified the incidence rates of HZ events per 100,000 person-years (PYs) of follow-up based on the total and subgroups of sex and 10-year age according to ICI treatment. To evaluate the incidence of HZ in patients with lung cancer relative to the general population between August 2017 and December 2021, we obtained sex-, five-year age-, and year-specific HZ incidences in the general population from the Korean Statistical Information Service and estimated the SIR. The SIR was defined as the ratio between the observed and expected numbers of HZ, depending on treatment with ICIs. The expected number was determined by multiplying the PYs in the cohort by the incidence rate of HZ in the cancer population according to sex, age, and calendar year. Additionally, 95% confidence intervals (CIs) were calculated by assuming the number of HZ events followed a Poisson distribution.

To investigate the effects of ICI treatment compared to those without ICI treatment on the occurrence of HZ, we compared the probability of occurrence of HZ associated with ICIs using Kaplan–Meier plots. Differences were estimated using log-rank tests. We calculated the observation time of patients from the first prescription date of ICIs or non-ICIs to the end of the follow-up period (31 December 2021), the mortality date, or the HZ event date. Additionally, we presented the hazard ratios (HRs) and 95% CIs in a Cox proportional hazards model adjusted for sex, age, comorbidities, concomitant use of immunosuppressants, and corticosteroid use. We also conducted stratified analyses according to sex, age (<68 years and ≥68 years), diabetes, cardiovascular disease, chronic lung diseases, chronic kidney diseases, chronic liver diseases, and rheumatic diseases, except for risk factors for which there were insufficient events to estimate the association. All reported *p*-values were two-sided, and a *p*-value of <0.05 was considered as statistically significant. Data processing and statistical analyses were conducted using the R software, version 4.0.3 (R Project for Statistical Computing).

## 3. Results

### 3.1. Patient Disposition and Baseline Characteristics

During the study period, data of 88,375 adults (age > 18 years) with the ICD code C34 for lung cancer were included. After excluding data of patients based on the exclusion criteria, records of 51,021 individuals who received chemotherapeutic regimens for lung cancer under the coverage of the NHI System were selected. Most patients with cancer included in this study were males and elderly. Of these, 897 (1.8%) patients were included in the ICI group and the remaining 50,124 (98.2%) in the non-ICI group (Figure 1). In the ICI group, 126 (14.0%), 2 (0.2%), 309 (34.4%), and 460 (51.3%) patients received atezolizumab, durvalumab, nivolumab, and pembrolizumab, respectively. The prevalence of diabetes, cardiovascular disease, and chronic lung disease was significantly lower in the ICI group than that in the non-ICI group. The number of patients who received steroids was significantly higher in the non-ICI group than that in the ICI group. The baseline characteristics of the study population are presented in Table 1.

### 3.2. Incidence Rate and Standardized Incidence Ratio of HZ in the ICI and Non-ICI Groups

Table 2 presents HZ events, PYs, and the incidence of lung cancer according to ICI treatment. We observed 29 cases of HZ in 2395.45 PYs in the ICI group and 2233 cases of HZ for 119,537.78 PYs in the non-ICI group. In an age-specific classification, the incidence of HZ was highest among individuals between 70 and 79 years in the ICI group (1529.44, 95% CI: 882.22–2508.54) and 60–69 years in the non-ICI group (2099.38, 95% CI: 1968.17–2237.11). According to sex, male patients exposed to ICIs (1270.54, 95% CI: 849.45–1839.98) showed a higher incidence of HZ than that seen in females (1025.29, 95% CI: 480.92–1993.90), while the incidence was converse in the non-ICI group.

We estimated the SIR to compare the incidence of HZ in patients with lung cancer to that in the general population (Table 3). The SIR values ranged between 0 and 5.75 in the ICI group and 0.77–30.22 in the non-ICI group. In the ICI group, male patients between 70 and 79 years showed the highest incidence of HZ than that seen in the general population (5.75, 95% CI: 3.19–10.39), followed by that in the 60–69 years (5.37, 95% CI: 2.69–10.75) and 50–59 years (4.26, 95% CI: 1.07–17.05) age groups. Individuals more than 80 years of age showed a 1.98-fold higher incidence of HZ; however, the difference was not statistically significant (*p* = 0.99). We observed significantly greater SIR values among females in the 50–59 years (4.68, 95% CI: 1.17–18.72) and 60–69 years (3.99, 95% CI: 1.29–12.37) age groups. In the non-ICI group, the highest SIR values were seen in both males and females (19.45, 95% CI: 9.73–38.90) < 40 years. The incidence rates of HZ among patients with lung cancer were higher than those in the general population across all age groups except for the ≥80 years group. We observed a 0.77-fold change (95% CI: 0.61–0.98) in HZ incidence for males and a 0.91-fold change (0.62–1.35) for that in females. The ICI and non-ICI groups of patients with lung cancer exhibited higher SIRs for HZ when compared to that for the general population, with a tendency of decreasing SIR values with increasing age.

### 3.3. Comparison of HZ Incidence between ICI and Non-ICI Groups

The probability of HZ incidence observed in the Kaplan–Meier plot is shown in Figure 2. The cumulative events of HZ 800 and 1600 days from enrollment were 24 and 29 for the ICI group and 1967 and 2233 for the non-ICI group, respectively. The probability of HZ occurrence in patients who were not treated with ICIs was higher than that in patients in the ICI group; however, the difference between the groups was only slightly significant (*p* = 0.053).

### 3.4. Risk Factors for Development of HZ in Patients with Lung Cancer

Table 4 shows the HR values of HZ incidence associated with ICI treatment compared to those without ICI treatment in the unadjusted and adjusted Cox model analyses. With respect to satisfaction of the proportional hazards assumption by Schoenfeld residuals (*p* = 0.36) and graphical evaluation, ICIs were associated with a 31% lower risk of HZ (0–52%, *p* = 0.05) in the adjusted model. In sex-specific association, female patients in the ICI group had a lower risk of developing HZ than females in the non-ICI group in both the unadjusted (HR, 0.43; 95% CI: 0.19–0.96, *p* = 0.04) and adjusted (HR, 0.42; 95% CI: 0.19–0.94, *p* = 0.04) models, while non-significant associations were observed in male patients. Additionally, ICIs compared to non-ICIs were associated with a lower risk of HZ in patients < 68 years (HR, 0.58; 95% CI: 0.34–0.99, *p* = 0.05). With respect to comorbidity, a significant association between ICI treatment and HZ incidence was observed in patients without chronic liver disease (HR, 0.68; 95% CI: 0.47–0.99, *p* = 0.05) and rheumatic diseases (HR, 0.65; 95% CI: 0.44–0.97, *p* = 0.03).

## 4. Discussion

In the present study, we evaluated HZ incidence according to sex and age and the effects of ICIs on HZ events in a lung cancer cohort using a nationwide population-based database. The incidence rate and SIR of HZ in patients with lung cancer receiving cytotoxic chemotherapeutic agents and/or targeted therapy were higher than those in patients treated with ICIs; this disparity was more notable in younger patients. The probability of occurrence of HZ according to Kaplan–Meier plots was higher in patients from the non-ICI group than that in patients from the ICIs group, with a marginal degree of statistical significance (*p* = 0.053). We also confirmed that ICIs treatment was associated with a lower risk of HZ than treatment without ICIs (approximately 31%), and this association was stronger in females and in those <68 years in age.

A previous study indicated that around 7% of patients diagnosed with malignant melanoma in the United States who underwent treatment with ICIs experienced notable infectious complications [15]. Moreover, *Pneumocystis* pneumonia (PCP) and tuberculosis have been observed in patients treated with nivolumab (PD-1 antibody) [16,17]. According to our understanding, however, treatment with ICIs does not exhibit elevated susceptibility to infections compared with alternative therapeutic approaches in current large randomized clinical trials [18,19,20,21]. In patients receiving PD-1/PD-L1 inhibitors, randomized trials have not shown an increased risk of infection [20,21]. A systematic review and meta-analysis reported a lower incidence of all-grade infections [22], and a retrospective case series showed that chronic viral reactivation was not observed in patients with HIV, hepatitis B virus (HBV), or hepatitis C virus (HCV) infections treated with ICIs [23].

Normally, VZV reactivation requires disrupted T-cell-mediated VZV immunity [24]. The cytolytic effects of CD8^+^ T cells inhibits VZV reactivation mediated by the production of IFN-γ, and CD8^+^ T-cell depletion might be associated with increased VZV reactivation [25]. Additionally, CD4^+^ T cells trigger cytokine secretions of TNF-α, IFN-γ, and IL-2, along with memory B-cell stimulation and IgG-mediated B-cell proliferation [26]. Low VZV-specific CD4^+^ T-cell responses correlate with severe HZ and complications such as post-herpetic neuralgia [27].

During VZV reactivation, VZV-specific CD4^+^ T cells acquire elevated levels of the T-cell inhibitory markers CTLA-4 and PD-1 [25,28]. One study [28] showed that PD-1 expression also increases in CD8^+^ T cells from HZ patients when compared to healthy controls. VZV infection may induce the expression of PD-1 in PBMCs, NK cells, and CD4^+^ and CD8^+^ T cells, potentially enhancing the capacity of infected immune cells to transmit the virus and impairing viral immune clearance efficacy. Another in vitro study [29] identified that immune suppression mediated by VZV can be reversed by enhancing CD8^+^ T-cell effector function by blocking PD-L1, thereby reducing virus dissemination and widespread disease. Thus, the function of VZV-specific CD4^+^ and CD8^+^ T cells may be enhanced upon ICI treatment by blocking inhibitory pathways and might protect against VZV reactivation. The induction of the PD-1/PD-L1 pathway during other viral infections has also been documented. Herpes simplex virus 1, another member of the alpha-Herpesviridae subfamily, induces PD-1 expression during infection, and the inhibition of PD-L1 in mice enhanced primary and secondary CD8^+^ T-cell immune responses [30].

Nevertheless, there is mounting evidence of several mechanisms for the development of infections associated with the administration of ICIs [31]. Immunosuppression associated with immune checkpoint-related leukopenia might lead to opportunistic infections [32]. However, mild to severe leukopenia associated with the PD-1/PD-L1 blockade only occurred in 0.9% of patients, indicating a less prevalent role for this mechanism [32]. Another hypothesis is that ICIs have the potential to trigger immune recovery, thereby facilitating the emergence of immune reconstitution inflammatory syndrome (IRIS) and the reactivation of latent/chronic infectious diseases with latency. Case reports in which the development of tuberculosis, [16,33] VZV-induced encephalitis [11], and pulmonary nocardiosis [34] in patients receiving PD-1/PD-L1 inhibitors without immunosuppressive treatment considered the possibility that IRIS might have been involved. Studies with animal models demonstrated that PD-1-deficient mice were found to be more susceptible to mycobacterial infection compared to wild-type mice, and they showed more evidence of fulminant infectious processes [35,36] with tissue damage caused by the overproduction of IFN-γ in effector T cells [37]. In HIV patients, VZV-IRIS has been associated with an increase in circulating CD8^+^ T cells after 1 month on ART or 1 month before the onset of HZ [38]. However, whether HZ susceptibility in patients with solid cancer could be clinically increased by dysregulated immunity due to ICIs has not yet been clarified. The analysis of T-cell subsets (CD4^+^, CD8^+^, regulatory T cells, etc.) at baseline and at the onset of HZ in patients receiving ICIs treatment may be useful in elucidating the mechanism underlying HZ incidence.

Notably, this observation does not diminish the importance of HZ prevention in lung cancer patients undergoing ICI monotherapy. Patients undergoing cancer treatment have been reported to have a higher incidence and severe symptoms of HZ than the general population [7,8]. In this study, the group receiving only ICIs also exhibited a high SIR for HZ compared with that seen in the general population across all age groups (Table 3). Additionally, autoimmune phenomena known as immune-related adverse events (irAEs) induced by PD-1/PD-L1 inhibitors may require treatment with immunosuppressive agents such as corticosteroids or antitumor necrosis factor agents, and these could contribute to the development of HZ or other opportunistic infections in this patient population [39]. Currently, there is a consensus recommending prophylaxis against HZ in patients receiving immunosuppressive regimens due to immunotherapy-related toxicities [40]. Moreover, misdiagnosis of infections may result in delayed diagnosis and treatment, as well as deterioration of the infectious condition due to the administration of corticosteroids and other immunosuppressants used for managing suspected irAEs. Hence, distinguishing between irAEs and infections is crucial. Substantial data similar to the findings of this study are required to serve as a foundation for understanding the immunobiology of various infectious diseases.

This is the first study to investigate the effects of ICIs on the risk of HZ in patients with lung cancer in South Korea. We used data on a nationwide population from the HIRA health insurance claims database. A similar study investigating the association between the rate of HZ and ICI treatment was conducted in Japan; however, it cannot be considered representative because of the small number of participants (n = 436) included in the study [14]. Multivariate analysis was conducted to control potential confounders after adjusting for a range of variables in the risk factor analysis. Finally, the SIR was accurately calculated based on the annual statistics of reported HZ cases and the general population census data published by the Korean government. The propensity matching method could be considered to adjust the skewness between groups, but the model using the matched dataset could not achieve convergence because the number of HZ events was too small (Appendix A).

However, this study had several limitations that must be considered when interpreting the results. First, owing to the observational nature of the claims database, HZ cases may have been underestimated or misclassified, and the severity of HZ was not assessed. Second, the dosage of ICIs was not considered. We classified the study population based on ICI exposure status because confirming the defined daily dose (DDD) or treatment duration is challenging due to the characteristics of ICIs administration regimens, and there is a lack of precise references regarding the dose-dependent impact of ICIs on immune response. A larger prospective observation study is needed to investigate the changes in immune modulation according to the dosage of ICIs administered. Third, there was a large difference in the number of patients in the ICIs and non-ICIs groups, and effects of only a limited number of ICIs (atezolizumab, durvalumab, nivolumab, and pembrolizumab), mostly pembrolizumab, were evaluated. And the number of ICIs prescriptions was too small to identify each stratified effect. Consequently, it was impossible to ascertain the differential effects of ICIs on the risk of developing HZ. Fourth, our assessment was limited specifically to lung cancer. In South Korea, ICIs treatment is covered by insurance for lung cancer, hepatocellular carcinoma, bladder cancer, melanoma, and renal cell cancer. Among these, cancers treated with cytotoxic chemotherapeutic agents only include lung and bladder cancers. Targeted therapies are also used for other diseases, resulting in fewer immunosuppressive side effects and a lower incidence of infectious diseases. Therefore, the target disease groups that aligned with the analytical objectives of this study were lung and bladder cancers. Considering the frequency of the disease, as the rate of lung cancer is significantly high, this study focused on patients with lung cancer. Further investigations are warranted to elucidate the risk of HZ in patients with different cancer types and those receiving treatment with other types of ICIs such as anti-CTLA-4, anti-LAG-3, and anti-TIM-3 inhibitors. Fifth, potential confounding variables including smoking/drinking habits, socioeconomic status, NSCLC subtype, and the impact of prior cytotoxic agents before ICI treatment were not adjusted for due to limitations in claims data availability. Moreover, clinical and laboratory data could not be adjusted as the HIRA database does not provide this information for individual claims. Sixth, the HZ vaccination status of patients could not be ascertained in this study. However, during the study period, the only available vaccines for HZ were all live attenuated vaccines (Zostavax^®^ [Sanofi Pasteur MSD, Lyon, France]) and the recombinant HZ vaccine (Shingrix^®^ [GSK, Rixensart, Belgium]) officially released in December 2022 in South Korea. Therefore, it can be speculated that the HZ vaccination rate among cancer patients may not have been high because of concerns regarding live vaccines. Finally, we were unable to verify HIV infection status from patient records, which is one of the most significant risk factors for HZ. This is because the HIRA database provides claims data with concealed individual identification of patients with HIV. However, because the data concealed the overall patient population, we believe that this did not have a significant impact on the findings of this study.

## 5. Conclusions

In summary, among patients with lung cancer, the risk of HZ was lower in the ICI group than that in the non-ICI group. As the use of ICIs is expected to gradually increase in the treatment of various cancers in the future, these challenges must be resolved by large prospective observational studies in order to maximize the clinical benefits of immunotherapy uninterruptedly and safely.

## Figures and Tables

**Figure 1 cancers-16-01499-f001:**
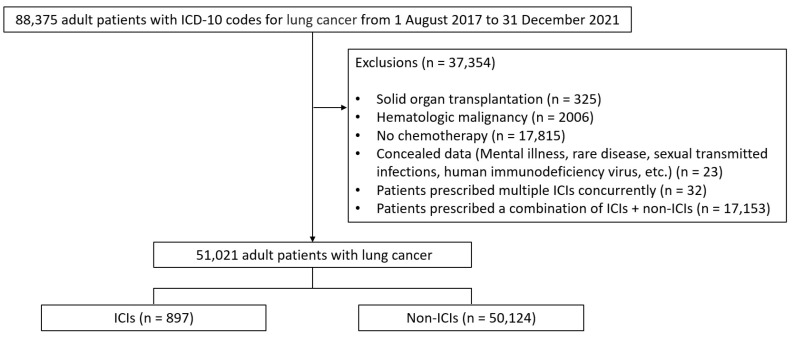
Flow chart showing details of the study population. ICD-10, International Classification of Diseases-10; HIV, human immunodeficiency virus; ICIs, immune checkpoint inhibitors.

**Figure 2 cancers-16-01499-f002:**
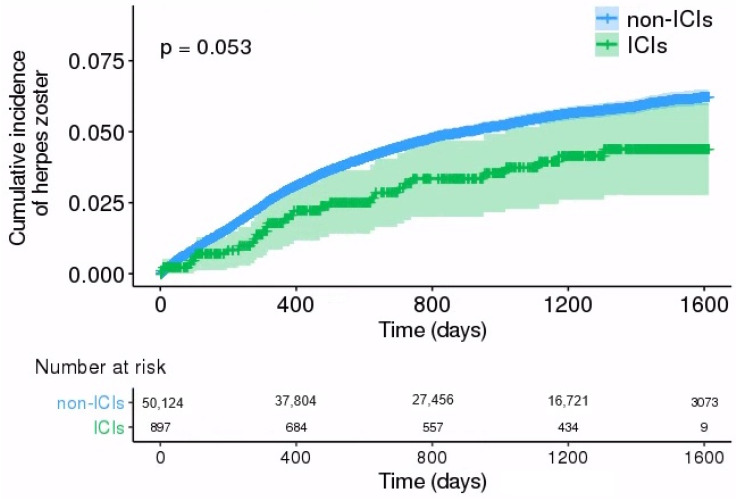
Kaplan–Meier curves showing the timeframe (in days) from study enrollment to events of herpes zoster (HZ) in the study population upon cancer treatment.

**Table 1 cancers-16-01499-t001:** Baseline characteristics of patients with lung cancer treated with immune checkpoint inhibitors (ICIs) or comparator non-ICIs.

	ICIs					Non-ICIs (n = 50,124)	*p*-Value
	Atezolizumab (n = 126)	Durvalumab (n = 2)	Nivolumab (n = 309)	Pembrolizumab (n = 460)	Total (n = 897)		
Age (year), n (%)							0.008
<40 (%)	1 (0.8)	0 (0)	1 (0.3)	4 (0.9)	6 (0.7)	475 (0.9)	
40–50 (%)	10 (7.9)	0 (0)	18 (5.8)	16 (3.5)	44 (4.9)	2019 (4.0)	
50–60 (%)	22 (17.5)	0 (0)	55 (17.8)	79 (17.2)	156 (17.4)	8040 (16.0)	
60–70 (%)	42 (33.3)	2 (100)	116 (37.5)	143 (31.1)	303 (33.8)	18,395 (36.7)	
70–80 (%)	43 (34.1)	0 (0)	96 (31.1)	159 (34.6)	298 (33.2)	17,555 (35.0)	
≥80 (%)	8 (6.3)	0 (0)	23 (7.4)	59 (12.8)	90 (10.0)	3640 (7.3)	
Sex, n (%)							0.3112
Male	89 (70.6)	2 (100)	253 (81.9)	334 (72.6)	678 (75.6)	37,109 (74.0)	
Female	37 (29.4)	0 (0)	56 (18.1)	126 (27.4)	219 (24.4)	13,015 (26.0)	
Comorbidity, n (%)							
Diabetes	51 (40.5)	0 (0)	106 (34.3)	193 (42.0)	350 (39.0)	21,708 (43.3)	0.011
Cardiovascular disease *	51 (40.5)	0 (0)	81 (26.2)	169 (36.7)	301 (33.6)	20,791 (41.5)	<0.001
Chronic lung diseases	94 (74.6)	1 (50)	199 (64.4)	307 (66.7)	601 (67.0)	36,234 (72.3)	0.001
Chronic kidney diseases	12 (9.5)	0 (0)	18 (5.8)	29 (6.3)	59 (6.6)	2751 (5.5)	0.179
Chronic liver diseases	6 (4.8)	0 (0)	10 (3.2)	9 (2.0)	25 (2.8)	1404 (2.8)	1.000
Rheumatic diseases	9 (7.1)	0 (0)	7 (2.3)	32 (7.0)	48 (5.4)	3221(6.4)	0.217
Concomitant use of immunosuppressive drugs, n (%)							
Immunosuppressant	10 (7.9)	0 (0)	6 (1.9)	15 (3.3)	31 (3.5)	2428 (4.8)	0.065
Steroid **	4 (3.2)	0 (0)	12 (3.9)	18 (3.9)	34 (3.8)	3386 (6.8)	0.001

* Includes ischemic heart disease, cerebrovascular disease, and diseases of the arteries, arterioles, and capillaries. ** Includes the use of corticosteroids defined as the presence of prescription records for prednisone equivalents ≥ 15 mg/day for at least 14 days.

**Table 2 cancers-16-01499-t002:** Incidence of herpes zoster corresponding to treatment with immune checkpoint inhibitors (ICIs).

	ICIs			Non-ICIs		
	Event (n)	Person-Years	Incidence * (95% CI)	Event (n)	Person-Years	Incidence * (95% CI)
Total	29	2395.45	1210.63 (844.97–1689.36)	2233	119,537.78	1868.03 (1791.34–1945.44)
Age, n (%)						
<40	0	13.92	0 (0–0)	19	1166.73	1628.48 (1047.07–2438.24)
40–49	0	131.22	0 (0–0)	74	5042.06	1467.65 (1170.00–1820.42)
50–59	4	446.80	895.25 (363.36–1962.22)	410	20,084.68	2041.36 (1853.26–2243.62)
60–69	11	849.69	1294.60 (729.75–2164.36)	921	43,870.17	2099.38 (1968.17–2237.11)
70–79	12	784.60	1529.44 (882.22–2508.54)	716	41,251.46	1735.70 (1610.87–1862.60)
≥80	2	169.22	1181.90 (365.61–3292.58)	93	8122.67	1144.94 (935.18–1389.07)
Sex, n (%)						
Male	23	1810.25	1270.54 (849.45–1839.98)	1471	89,520.82	1643.19 (1561.37–1728.22)
Female	6	585.20	1025.29 (480.92–1993.90)	762	30,016.96	2538.56 (2364.71–2721.94)

* Incidence per 100,000 person-years (PYs). CI, confidence interval.

**Table 3 cancers-16-01499-t003:** Standardized incidence ratios (SIRs) of herpes zoster for the general population with 95% confidence intervals (CIs), according to treatment, age, and sex.

	ICIs				Non-ICIs			
	Observed Event (n)	Expected Event (n)	SIR (95% CI)	*p*-Value	Observed Event (n)	Expected Event (n)	SIR (95% CI)	*p*-Value
Male								
<40	0	0.01	0 (0–0)	1.00	11	0.36	30.22 (16.74–54.57)	<0.01
40–49	0	0.10	0 (0–0)	1.00	37	3.16	11.69 (8.47–16.13)	<0.01
50–59	2	0.47	4.26 (1.07–17.05)	0.04	224	21.01	10.66 (9.35–12.15)	<0.01
60–69	18	1.49	5.37 (2.69–10.75)	<0.01	609	78.64	7.74 (7.15–8.38)	<0.01
70–79	11	1.91	5.75 (3.19–10.39)	<0.01	521	98.19	5.32 (4.88–5.79)	<0.01
≥80	2	1.98	1.01 (0.25–4.04)	0.99	68	88.02	0.77 (0.61–0.98)	0.03
Female								
<40	0	0	0 (0–0)	1.00	8	0.41	19.45 (9.73–38.90)	<0.01
40–49	0	0.08	0 (0–0)	1.00	37	4.12	8.99 (6.51–12.40)	<0.01
50–59	2	0.43	4.68 (1.17–18.72)	0.03	186	19.34	9.62 (8.33–11.10)	<0.01
60–69	3	0.75	3.99 (1.29–12.37)	0.02	312	36.09	8.65 (7.74–9.66)	<0.01
70–79	1	0.51	1.96 (0.28–13.9)	0.50	194	29.60	6.55 (5.69–7.54)	<0.01
≥80	0	0.43	0 (0–0)	1.00	25	27.43	0.91 (0.62–1.35)	0.64

**Table 4 cancers-16-01499-t004:** Hazard ratio (HR) values of herpes zoster incidence associated with immune checkpoint inhibitors (ICIs) compared to that for non-ICIs.

	Unadjusted		Adjusted	
	HR (95% CI)	*p*-Value	HR (95% CI)	*p*-Value
Total	0.70 (0.48–1.01)	0.05	0.69 (0.48–1.00)	0.05
Sex				
Male	0.84 (0.55–1.26)	0.39	0.84 (0.55–1.26)	0.40
Female	0.43 (0.19–0.96)	0.04	0.42 (0.19–0.94)	0.04
Age				
<68 years	0.59 (0.35–0.99)	0.05	0.58 (0.34–0.99)	0.05
≥68 years	0.84 (0.51–1.40)	0.51	0.84 (0.50–1.40)	0.50
Diabetes				
No	0.81 (0.53–1.23)	0.32	0.84 (0.55–1.28)	0.41
Yes	0.49 (0.23–1.03)	0.06	0.48 (0.23–1.01)	0.05
Cardiovascular disease				
No	0.65 (0.42–1.01)	0.06	0.65 (0.41–1.01)	0.05
Yes	0.80 (0.42–1.55)	0.52	0.85 (0.44–1.64)	0.63
Chronic lung diseases				
No	0.59 (0.32–1.11)	0.10	0.58 (0.31–1.09)	0.09
Yes	0.76 (0.48–1.19)	0.23	0.76 (0.48–1.20)	0.24
Chronic kidney diseases				
No	0.71 (0.49–1.03)	0.07	0.70 (0.48–1.02)	0.07
Yes	0.52 (0.07–3.74)	0.52	0.52 (0.07–3.89)	0.53
Chronic liver diseases				
No	0.68 (0.47–0.99)	0.05	0.68 (0.47–0.99)	0.05
Yes	1.57 (0.22–11.41)	0.65	1.18 (0.13–10.91)	0.88
Rheumatic diseases				
No	0.66 (0.45–0.97)	0.03	0.65 (0.44–0.97)	0.03
Yes	1.52 (0.48–4.78)	0.47	1.76 (0.52–5.93)	0.36

CI, confidence interval.

## Data Availability

Data presented in this study are available upon request from the corresponding author.

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
