# Peer review of "Reactivation of Varicella-Zoster Virus in Patients with Lung Cancer Receiving Immune Checkpoint Inhibitors: Retrospective Nationwide Population-Based Cohort Study from South Korea"

_cancers, 2024, doi:10.3390/cancers16081499_

Round 1
Reviewer 1 Report
Comments and Suggestions for Authors
From a biostats and clinical epidemiology point of view, here are some comments for the Authors.
- line 21, HZV rather than HZ (everywhere along the manuscript)
- line 24 prescription of ICIs at least once during the study period, well we can't really consider that 1 IT cycle is a true IT therapy, moreover due to the well known phenomenon of pseudoprogression! I do warmly suggest to totally reclassify the study population for IT (i.e. fixing a minumum number of cycle as cutoff for IT yes/no)
- line 29, the huge prevalence of HZV among males should be better clarified
- line 32, KM curves have not to be applied to estimate incidence (mind that their outcome is a time-to-event); when dealing with competing events, the ideal model is the estimation of cumulative incidence (Fine & Gray model)
- line 33 treatment with ICIs was associated with a 31% lower risk of developing HZV, this is an hard statement and it must be justified in details; in a way, you are stating that IT administration protects from HZV infections, hard to be accepted in toto (since usually the opposite is true)
- line 116, median follow-up, overall and stratified by IT use, is lacking
- line 128 Corticosteroid use was defined as the presence of prescription records for prednisone equivalents ≥ 15 mg/day for at least 14 days, mind that a quite lesser and shorter administration of steroids may deep interphere with HZV reactivation
- line 143, age, as any continuous covariate, has to be reported as median/IQR and quantitatively estimated rather than qualitatively (i.e. per 10-yr classes)
- line 157 we compared the probability of occurrence of HZ associated with ICIs using Kaplan–Meier plots, this is a wrong concept since you are using a time-to-event model, remember that you are estimating a time not an incidence!
- line 174 897 (1.8%) patients were included in the ICIs group and the remaining 50,124 (98.2%) in the non-ICIs group, mind the extreme asymmetrical distribution of your cohort, affecting model stability
- line 185, the previous lines of CT administered just before IT (a lacking info), can deeply affect HZV reactivation, this topic has to be investigated and mentioned
- line 219 and figure 2, this is a serious flaw, you are not dealing with cumulative incidence since no competing risks have been defined and estimated; moreover, cumulative incidence and competing risks regression (expressed as SDHR and not HR) are the correct modeling tools, while KM and Cox regression can not be used in this context
- line 356 the incidence of HZ was lower in the ICIs group than that in the non-ICIs group, this is to be proved de novo with an appropriate modeling approach, being the current one inappropriate
Comments on the Quality of English Languageminor
Reviewer 2 Report
Comments and Suggestions for Authors
In this retrospective study, the authors assessed the risk of Herpes Zoster in lung cancers in patients who received or not Immune Checkpoint Inhibitors. The authors reported that the rate of HZ reduced with ICI inihibtors. The authors assessed the risk factors associted with HZ development and they found no significance with any of the risk factors (Diabetes, Cardiovascular diseases,..etc)
Although the authors included a larger cohort, there are many missing points
1) THe doses and durations of ICIs are not mentioned.
2) The method of HZ diagnosis (serology, molecular) and clinical symptoms are not mentioned.
3) The duration of diseases (HZ) is missing.
4) The authors focused on demographic data and risk factors, while the clinical data and laboratory data such as blood pictures are poor.
5) The stage of lung cancers should be added in both groups
Comments on the Quality of English LanguageModerate
Reviewer 3 Report
Comments and Suggestions for Authors
I was invited to revise the paper entitled "Reactivation of Varicella-Zoster Virus in Patients with Lung Cancer Receiving Immune Checkpoint Inhibitors: Retrospective Nationwide Population-Based Cohort Study of the Health Insurance Review and Assessment Database in South Korea". It was a cohort study aimed to evaluate the incidence of HZ among patients with lung cancer treated with immunce checkpoint inhibitors, compared to paients treated with citotoxic drugs.
THe topic is very interesting a focused on an important topic for public health.
Observations:
- The title appears too long. I suggest to reduce it. I propose: Reactivation of Varicella-Zoster Virus in Patients with Lung Cancer Receiving Immune Checkpoint Inhibitors: Retrospective Nationwide Population-Based Cohort Study from South Korea;
- Methods are deeply described. Due to baseline differences in comorbidities between study groups, is suggest to perform a matching procedure in order to obtain homogenous groups (propensity score?);
- As subanalysis, I suggest to perform a comparison also among specific ICP-I drugs;
- Among limitations, I suggest to add also the lack of information about chickenpox diseases. Patients without history of varicella cannot be considered at risk for HZ.
Reviewer 4 Report
Comments and Suggestions for Authors
This is a retrospective study that looks backwards and examines exposures to immune checkpoint inhibitors in lung cancers, in relation to the outcome of reactivation of varicella-zoster virus. Outcome that is established at the start of the study. This type of study has limitations. Most sources of error are due to confounding and bias. In retrospective studies the odds ratio provides an estimate of relative risk (this study is using Hazard ratio values, both unadjusted and adjusted).
(1) Could you please explain how the design of the study dealt with confounding variables?
(2) How was bias assessed?
(3) Did you make any propensity score matching?
(4) Why Odds Ratio was not calculated?
(5) How was the reactivation of HZ evaluated? What were the clinical criteria?
(6) Are both ICI and non-ICI groups comparable?
(7) There are several types of immune checkpoint inhibitors (ICI)? Should the analysis be made for each type of ICI?
(8) What is the result if you calculate a Cox regression analysis, univariate and multivariate. The input variables would be ICI and the other clinicopathological variables of the patients.
(9) ICI was associated to a less incidence of HZ.
(10) Lung cancer is a broad group, there are several histological subtypes. Were the subtypes analyzed?
(11) The clinical characteristics are shown in Table 1. Apart from age, sex, and comorbidities, what about the other relevant clinical variables of the patients of lung cancer such as stage?
(12) Is the immune suppression of >15 mg/day for at least 14 days enough to induce/facilitate HZ reactivation?
(13) What is the relationship of the use of steroides and ICI. Non-ICI used more steroids, which supress immune system, and may facilitate HZ reactivation. Could this be the cause?
(14) In figure2, p value is 0.053. Therefore, in general, this is a negative result? Could some of the comorbidities be associated to a higher risk of HZ?
(15) Please discuss limitations at the end of the discussion.
(16) A more detailed description, including figures, of HZ and ICI mechanisms would enhance quality of the manuscript.
Reviewer 5 Report
Comments and Suggestions for Authors
Thank you for granting me the opportunity to review the article titled "Reactivation of Varicella-Zoster Virus in Patients with Lung Cancer Receiving Immune Checkpoint Inhibitors: Retrospective Nationwide Population-Based Cohort Study of the Health Insurance Review and Assessment Database in South Korea" (cancers-2877934). The manuscript is submitted to the "Cancer Immunology and Immunotherapy" section of the Special Issue on "Immunotherapy for Cancers."
This article, encompassing a considerable sample size of 51,021 study participants, observed data between August 2017 and December 2021. Its primary objective was to investigate the correlation between immune checkpoint inhibitors (ICIs) and the incidence of herpes zoster (HZ) in patients diagnosed with lung cancer.
The title is both informative and precise, aligning seamlessly with the content of the study. The abstract is meticulously structured, effectively conveying the key information of the research. While the reintroduction is well-articulated and substantiates the research hypothesis, I suggest considering positioning the research objective as the concluding segment of the introduction. Additionally, it may be beneficial to present the methodology as a hypothesis earlier in the introduction.
The methodology is lucidly outlined and specific. However, I propose that Table 1 be relocated from the methodology section to the results section. The baseline characteristics of the study population appear more aptly because of the authors' analysis.
Regarding the results, Figure 1 illustrating the selection of study participants should be incorporated into the materials and methods section rather than the results section.
In the discussion section, the authors offer a thoughtful reflection on the obtained results. I concur with the identified limitations and endorse the call for future prospective studies on this subject.
Round 2
Reviewer 1 Report
Comments and Suggestions for Authors
I'm extremely surprised to be reassigned once again the same manuscript I already rejected. Methodology approach does remain quite poor, all the KM curves are void (!), the concept of cumulative incidence has not been taken in account by the authors and so on
Comments on the Quality of English Languageminor
Author Response
Thank you very much for your insightful comments. I greatly appreciate your acknowledgment of that aspect, and I share the sentiment that it would have been highly desirable to include such analysis. However, regrettably, due to constraints posed by the current dataset, conducting such an analysis was challenging. It would be beneficial to gather more detailed and extensive data or prospective data for future research endeavors.
Reviewer 2 Report
Comments and Suggestions for Authors
No further comments
Comments on the Quality of English Languagefine
Author Response
We are appreciated for your generous comment.
Reviewer 3 Report
Comments and Suggestions for Authors
Authors addressed properly the great part of my previous observation.
About matching procedure, a PSmatcing can't led to selection bias. It allow to simulate a randomization leading to an appropriate covariates balance.
In my opinio, Authors should perform this analysis.
Reviewer 4 Report
Comments and Suggestions for Authors
Thank you for the answers. I do not have any additional comment.
Author Response

(The authors gave the same response as above.)

Round 3
Reviewer 3 Report
Comments and Suggestions for Authors
I thanks Authors for the appropriate respond. I suggest to add psmatching table as supplementary material. The paper can now be accepted for publication.
Author Response
We have attached the mentioned content as supplementary material. We want to express our sincere gratitude for your thorough review of my manuscript and for providing constructive feedback that significantly contributed to its improvement.